# A compelling symmetry: The extended fetuses-at-risk perspective on modal, optimal and relative birthweight and gestational age

## K. S. Joseph *

Department of Obstetrics and Gynaecology, School of Population and Public Health, University of British Columbia and the Children's and Women's Hospital and Health Centre of British Columbia, Kelowna, Canada

* kjoseph@cw.bc.ca

## Abstract

### Background

The relationship between several intriguing perinatal phenomena, namely, modal, optimal, and relative birthweight and gestational age, remains poorly understood, especially the mechanism by which relative birthweight and gestational age resolve the paradox of intersecting perinatal mortality curves.

### Methods

Birthweight and gestational age distributions and birthweight- and gestational age-specific perinatal death rates of low- and high-risk cohorts in the United States, 2004–2015, were estimated using births-based and extended fetuses-at-risk formulations. The relationships between these births-based distributions and rates, and the first derivatives of fetuses-at-risk birth and perinatal death rates were examined in order to assess how the rate of change in fetuses-at-risk rates affects gestational age distributions and births-based perinatal death rate patterns.

### Results

Modal gestational age typically exceeded optimal gestational age because both were influenced by the peak in the first derivative of the birth rate, while optimal gestational age was additionally influenced by the point at which the first derivative of the fetuses-at-risk perinatal death rate showed a sharp increase in late gestation. The clustering and correlation between modal and optimal gestational age within cohorts, the higher perinatal death rate at optimal gestational age among higher-risk cohorts, and the symmetric left-shift in births-based gestational age-specific perinatal death rates in higher-risk cohorts explained how relative gestational age resolved the paradox of intersecting perinatal mortality curves.

### Conclusions

Changes in the first derivative of the fetuses-at-risk birth and perinatal death rates underlie several births-based perinatal phenomena and this explanation further unifies the fetuses-at-risk and births-based models of perinatal death.

**Data Availability Statement:** The data are publicly available at https://www.cdc.gov/nchs/data_access/vitalstatsonline.htm.

**Funding:** KSJ's work is supported by an Investigator award from the BC Children's Hospital Research Institute. The funder had no role in study design, data collection and analysis, decision to publish, or preparation of the manuscript.

**Competing interests:** The author has declared that no competing interests exist.

## Introduction

Several studies have shown that population cohorts based on nationality, racial origin and other characteristics vary substantially in terms of birthweight distribution and optimal birthweight (i.e., the birthweight at which perinatal mortality rates are lowest) [1–9]. A related enigmatic finding is that optimal birthweight typically exceeds modal birthweight (i.e., the maximum of the birthweight distribution) [7–9]. Although it is unclear why many fetuses in diverse populations are born before reaching optimal size, these findings have led to recommendations regarding the need for population-specific standards of birthweight for identifying small infants at risk of perinatal death [8].

Some support for the proposition that perinatal mortality risk is best assessed through population-specific standards of birthweight is also forthcoming from the literature on the paradox of intersecting perinatal mortality curves. This phenomenon was first described over 50 years ago by Yerushalmy who showed that neonatal death rates favoured the low birthweight infants of mothers who smoked (compared with the low birthweight infants of mothers who did not smoke), while the opposite was true at higher birthweights [10]. The paradox is now recognized to be a general phenomenon [11–25] that is observed across numerous contrasts (e.g., infants of hypertensive vs normotensive mothers [14], and singletons vs twins [13,15,16]), outcomes (e.g., stillbirths and cerebral palsy [11–19]) and indices of prematurity (gestational age and birthweight [11–25]). One of the first attempts at resolving the paradox involved an intriguing reformulation involving relative birthweight and relative gestational age (i.e., with absolute birthweight or gestational age in each population recast in terms of its mean and standard deviation) [7,17]. When birthweight- and gestational age-specific perinatal death rates are quantified in terms of relative birthweight or relative gestational age, infants of mothers who smoke (have hypertension, etc) have higher rates of perinatal death at all birthweights and gestational ages [5–7,9,12,14,15,17,25–28].

A recent paper [29] offered evidence in favour of the proposition that the rate of change in the birth rate of a population (i.e., the first derivative of the population's fetuses-at-risk birth rate) determines the population's gestation age distribution, and that the first derivatives of the birth rate and the fetuses-at-risk perinatal mortality rate together determine the population's births-based gestational age-specific perinatal mortality pattern. This unifies the fetuses-at-risk and births-based models of perinatal death and also explains various perinatal phenomena including the early gestation exponential decline and the late gestation exponential increase in births-based perinatal mortality rates, and also the paradox of intersecting perinatal morality curves [29,30]. In this paper, the first derivatives of the birth rate and the fetuses-at-risk perinatal mortality rate are used to explain other previously unexplained phenomena, namely, modal, optimal and relative birthweight and gestational age. Understanding these phenomena, especially the mechanism by which relative gestational age uncrosses intersecting perinatal mortality curves, will provide further support for unifying the two models of perinatal death.

## Methods

### Background and rationale for the study

The seemingly opposed perspectives of the births-based and fetuses-at-risk models [29] can be reconciled by viewing the early gestation exponential decline in births-based perinatal death rates as being the product of an initially accelerating birth rate (i.e., steep increase in the first derivative of the fetuses-at-risk birth rate) and a fetuses-at-risk perinatal death rate that is stable or slowly accelerating in early gestation (no change or a small increase in the first derivative of the fetuses-at-risk perinatal death rate). Similarly, the late gestation increase in births-based

perinatal death rates can be explained as a product of a decelerating birth rate (i.e., sharp declines in the first derivative) and an abrupt acceleration in the fetuses-at-risk perinatal death rate (i.e., sharp increase in the first derivative). Births-based perinatal death rates fall exponentially in early gestation because the accelerating birth rate results in an increasing number of births, whereas the number of perinatal deaths is essentially unchanged as a consequence of the stable or slowly accelerating fetuses-at-risk perinatal death rate. On the other hand, the late gestation rise in births-based perinatal death rates occurs because reductions in acceleration (or a deceleration) in the birth rate at later gestation leads to a relatively smaller increase (or a fall) in the number of births, whereas the number of perinatal deaths rises sharply because of the rapidly accelerating fetuses-at-risk perinatal death rate [29,30]. Compared with low-risk cohorts, higher-risk cohorts show a steeper increase in the first derivative of the birth rate at early gestation (i.e., greater acceleration in the birth rate), and an earlier peak and an earlier decline in this first derivative at late gestation (i.e., earlier reductions in acceleration in the birth rate). The left-shift in the distribution of the first derivative of the birth rate in higher-risk cohorts is responsible for a left-shift in gestational age distributions and in births-based perinatal death rate curves. The latter left-shift in births-based perinatal death rates of higher-risk cohorts results in the paradox of intersecting perinatal mortality curves [29,30].

The rationale for the present study was premised on the above-mentioned propositions: if the rate of change in the birth rate determines the birth rate pattern and influences the gestational age distribution, and if the rate of change in the birth rate and the rate of change in the fetuses-at-risk perinatal death rate together influence the pattern of births-based gestational age- and birthweight-specific perinatal death rates, it is likely that the rate of change in fetuses-at-risk birth and perinatal death rates also underlie the phenomena of modal, optimal, and relative birthweight and relative gestational age. The rate of change in the birth rate is of particular interest as it's magnitude at specific points in gestation is not immediately evident from the exponentially increasing birth rate.

## Data source and analysis

All live births and stillbirths in the United States from 2004 to 2015 were included in the study with data obtained from the fetal death and period linked birth-infant death files of the National Center for Health Statistics. The study population was restricted to births with a clinical estimate of gestation between 20 and 43 weeks. Twelve low- and high-risk cohorts were identified, namely, singletons of women who did not have hypertension or diabetes (referred to as low-risk singletons), singletons of women with hypertension, singletons of women with diabetes, singletons of women with hypertension and diabetes, White singletons, Black singletons, singletons of women aged 25–29 years, singletons of women aged ≥35 years, singletons of women with a previous preterm birth, singletons of women without a previous preterm birth, twins, and triplets.

Preliminary examination of the birthweight distribution showed substantial ounce and digit preference in birthweight values (S1 Fig in S1 Appendix) and birthweight was therefore categorized into 28 g birthweight groups centred on the gram equivalent of each complete ounce. The birthweight distribution and its modal value, and the birthweight-specific perinatal death rate (including stillbirths and neonatal deaths) and its lowest point (i.e., optimal birthweight) were then estimated by fitting splines to the log transformed birthweight groups and birthweight-specific perinatal death rates using the Transreg procedure in the SAS statistical software package (SAS Institute, Cary, NC).

The frequency distribution of gestational age and gestational age-specific perinatal death rates were calculated under the births-based formulation (expressed per 1,000 total births) and

modal and optimal gestational age were estimated. Gestational age-specific birth rates and gestational age-specific fetuses-at-risk perinatal death rates (both expressed per 1,000 fetus-weeks) were also calculated using the extended fetuses-at-risk formulation [28,31–36]. The number of births (or perinatal deaths) at any gestational week constituted the numerator for these fetuses-at-risk rates, while the fetal-time accrued by the fetuses at risk over the gestational week in question constituted the denominator. Fetal-time was estimated by averaging the number of fetuses at the beginning and the end of the gestational week of interest (which included fetuses delivered at that gestational week and those delivered subsequently; S1 and S2 Tables in S1 Appendix).

The Expand procedure in the SAS statistical package was used to estimate the first derivatives of the fetuses-at-risk gestational age-specific birth rates and the fetuses-at-risk gestational age-specific perinatal death rates (S3 Table in S1 Appendix). The first derivatives were computed from cubic splines fitted to the fetuses-at-risk birth and perinatal death rates and quantified the *rate of change* (increase or decrease) in these rates at each gestational week. It may be helpful to view the birth rate (births per 1,000 fetus-weeks) and its first derivative (births per 1,000 fetus-weeks per week, or births per 1,000 fetus-weeks$^2$) as being analogous to velocity (metres/sec) and acceleration/deceleration (metres per second per second, or metres per second$^2$), respectively. Thus, a positive first derivative of the birth rate represents an accelerating birth rate while a negative first derivative represents a decelerating birth rate. A positive and continually increasing first derivative of the birth rate signifies a progressively increasing acceleration in the birth rate, while a positive and progressively decreasing first derivative signifies a birth rate that is increasing but at a slower rate (i.e., with reduced acceleration) than in previous gestational weeks.

Birthweight and gestational age distributions, gestational age-specific birth rates, the derivatives of the birth rates, births-based and fetuses-at-risk perinatal death rates, and the derivatives of the fetuses-at-risk perinatal death rates were estimated for each low- and high-risk cohort and graphed in order to examine potential relationships with modal, optimal and relative birthweight and gestational age (i.e., with the latter calculated using z-scores based on the mean and standard deviation of the birthweight and gestational age distributions of each cohort). Correlations between the gestational age at which the first derivative of the birth rate peaked and the mean, mode, median and optimal birthweight and gestational age were estimated in the 12 cohorts using Pearson correlation coefficients (r). Correlations between the gestational age at which the first derivative of the fetuses-at-risk perinatal death rate showed an abrupt upward increase at late gestation and optimal birthweight and optimal gestational age were similarly assessed.

All analyses were based on anonymized, publicly available data and ethics approval for the study was not sought.

## Results

There were 47,626,172 live births and stillbirths between 20 and 43 weeks' gestation in the study population. The rate of perinatal death varied substantially between the different cohorts; it was 8.2 per 1,000 total births among low-risk singletons, and 72.4 per 1,000 total births among triplets (S4 Table in S1 Appendix).

Fig 1A and 1B shows birthweight distributions, birthweight-specific perinatal death rates and modal and optimal birthweight among low-risk singletons and twins. Modal birthweight was substantially lower than optimal birthweight in both cohorts, and modal birthweight and optimal birthweight were substantially lower among twins; similarly, modal and optimal gestational age were lower among twins (37 and 38 weeks, respectively) compared with low-risk

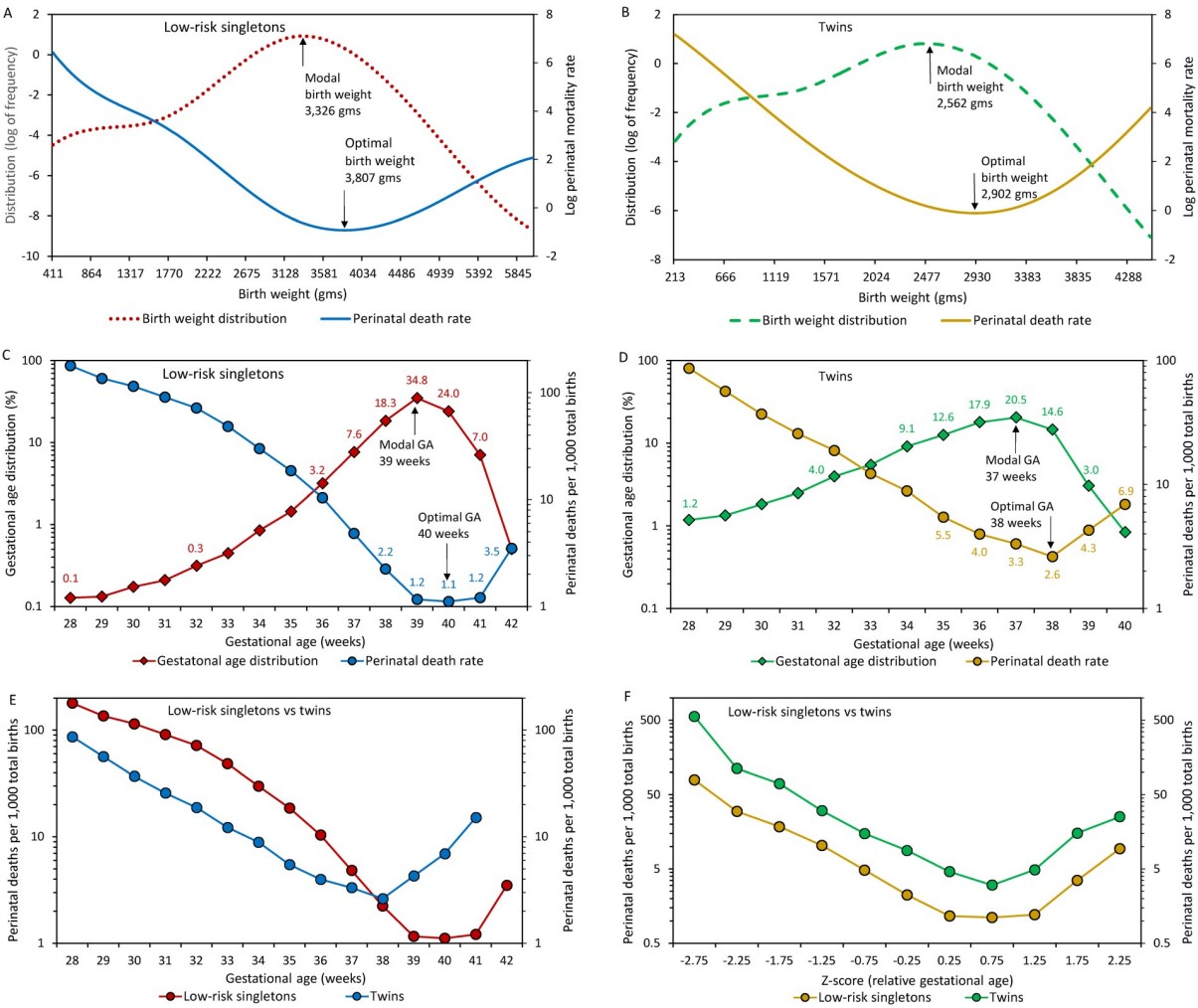

**Fig 1.** Birthweight distributions and birthweight-specific perinatal death rates among singletons of low-risk women (i.e., without hypertension or diabetes; Panel A) and twins (Panel B); gestational age distributions and gestational age-specific perinatal death rates among singletons of low-risk women (Panel C) and twins (Panel D); and births-based gestational age-specific perinatal death rates (Panel E) and births-based relative gestational age-specific perinatal death rates (Panel F) among singletons of low-risk women and twins, United States, 2004–2015.

singletons (39 weeks and 40 weeks, respectively; Fig 1C and 1D). The lowest gestational age-specific perinatal death rate among twins was higher than the lowest perinatal death rate among low-risk singletons. The births-based perinatal death rate curves of the two cohorts intersected; perinatal death rates were lower among twins <38 weeks' and higher at 38 weeks' gestation and over compared with perinatal death rates among low-risk singletons (Fig 1E). When gestational age-specific perinatal death rates were based on relative gestational age (z-scores), twins had higher rates of perinatal death at all gestational ages (Fig 1F).

Fig 2 shows the birth rate, the rate of change in the birth rate and the gestational age distribution among the singletons of low-risk women and twins. The first derivative of the birth rate was left-shifted (Fig 2B), the birth rate was considerably higher at each gestational week (Fig 2A), and the gestational age distribution was substantially left-shifted among twins (Fig 2C).

Fig 3 shows the birth rates and their first derivatives, the fetuses-at-risk perinatal death rates and their derivatives and births-based perinatal death rates in the two cohorts. The first

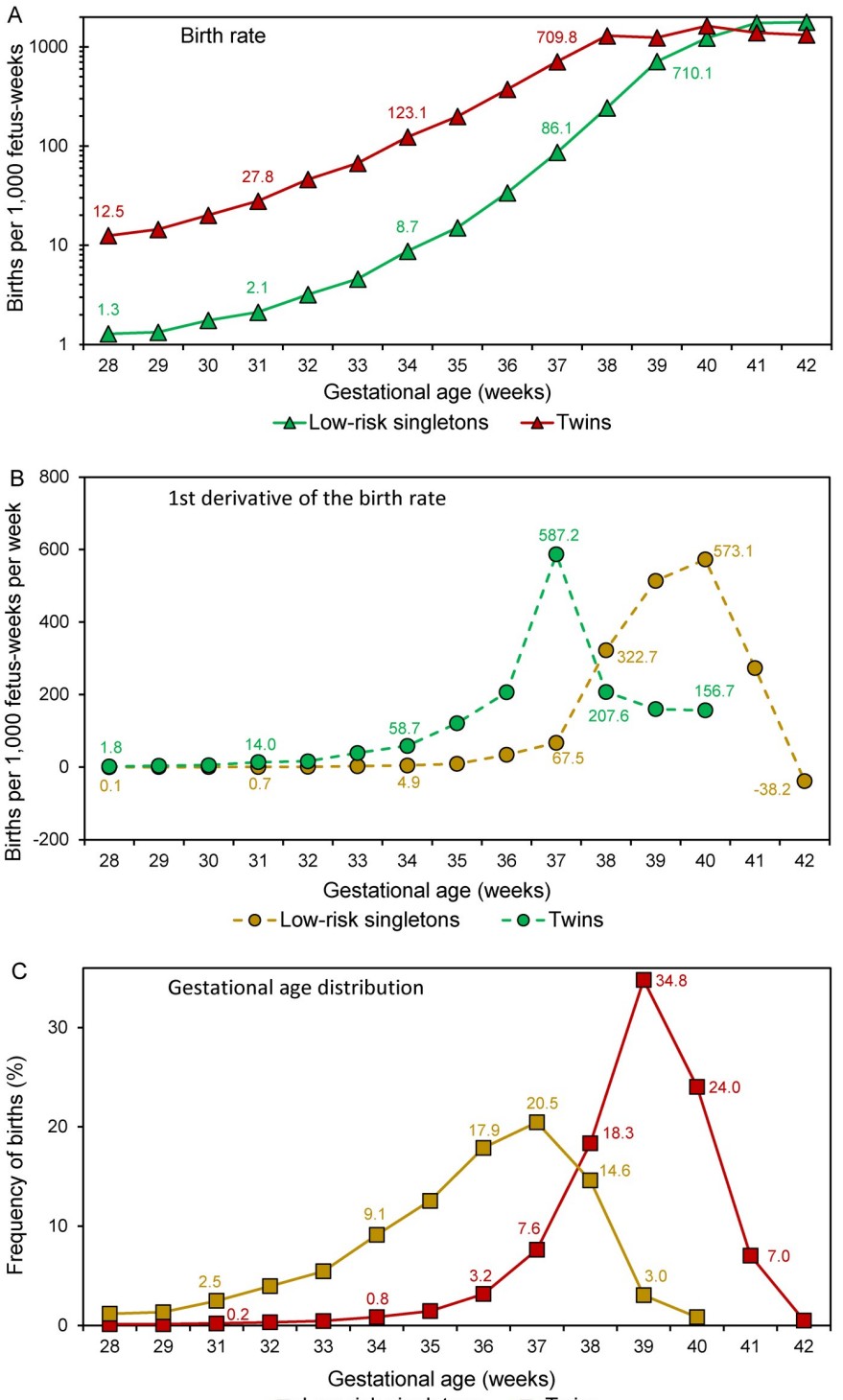

**Fig 2.** Gestational age-specific birth rates among singletons of low-risk women (i.e., without hypertension or diabetes) and twins (Panel A), the first derivative of the birth rate among singletons of low-risk women and twins (Panel B) and gestational age distributions (Panel C) among singletons of low-risk women and twins, United States, 2004–2015.

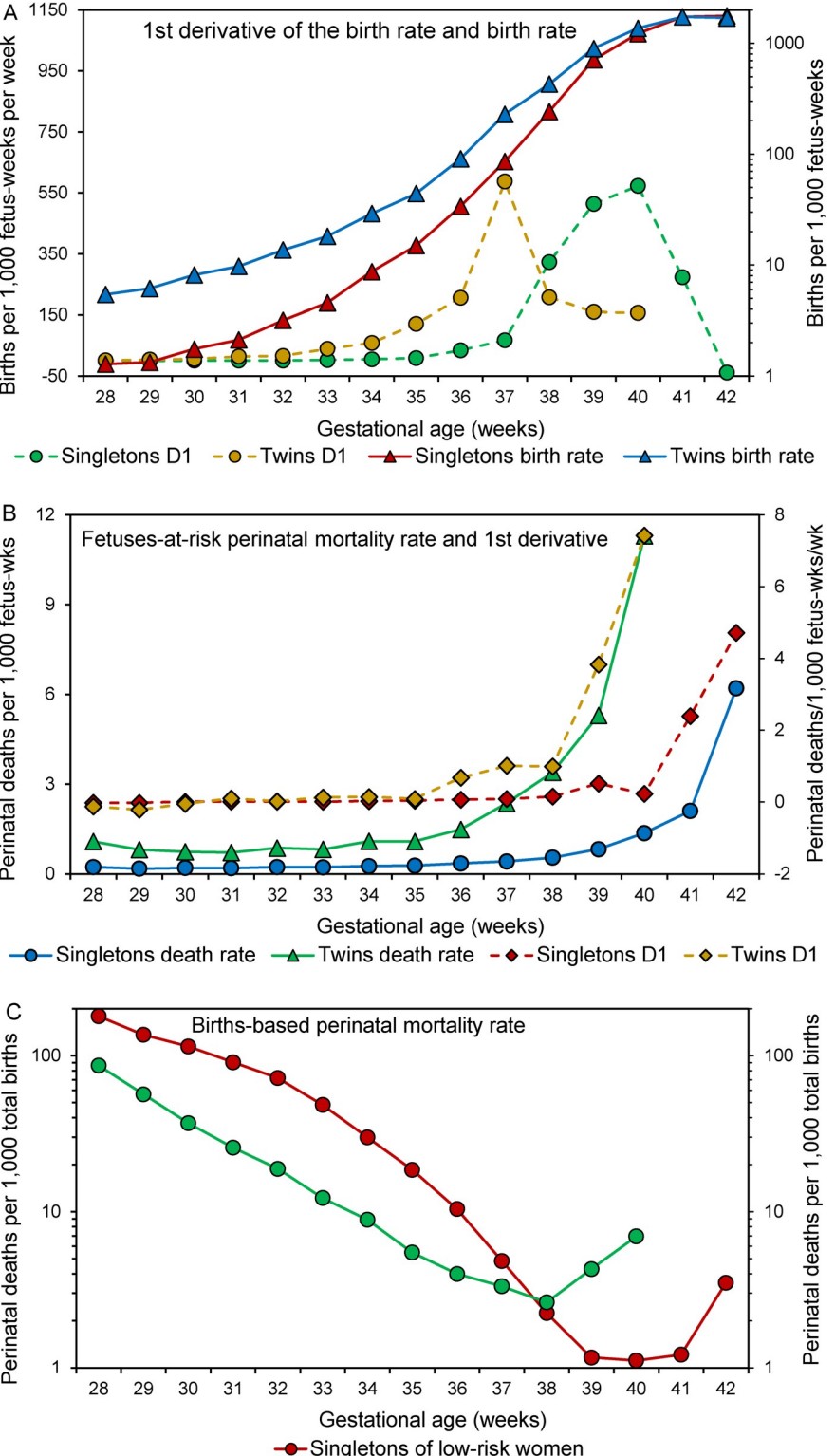

**Fig 3.** Gestational age-specific birth rates and their first derivatives among singletons of low-risk women (without hypertension or diabetes) and twins (Panel A), gestational age-specific fetuses-at-risk perinatal death rates and their first derivatives among singletons of low-risk women and twins (Panel B), and births-based gestational age-specific perinatal death rates (Panel C) among singletons of low-risk women and twins, United States, 2004–2015 (D1 denotes first derivative).

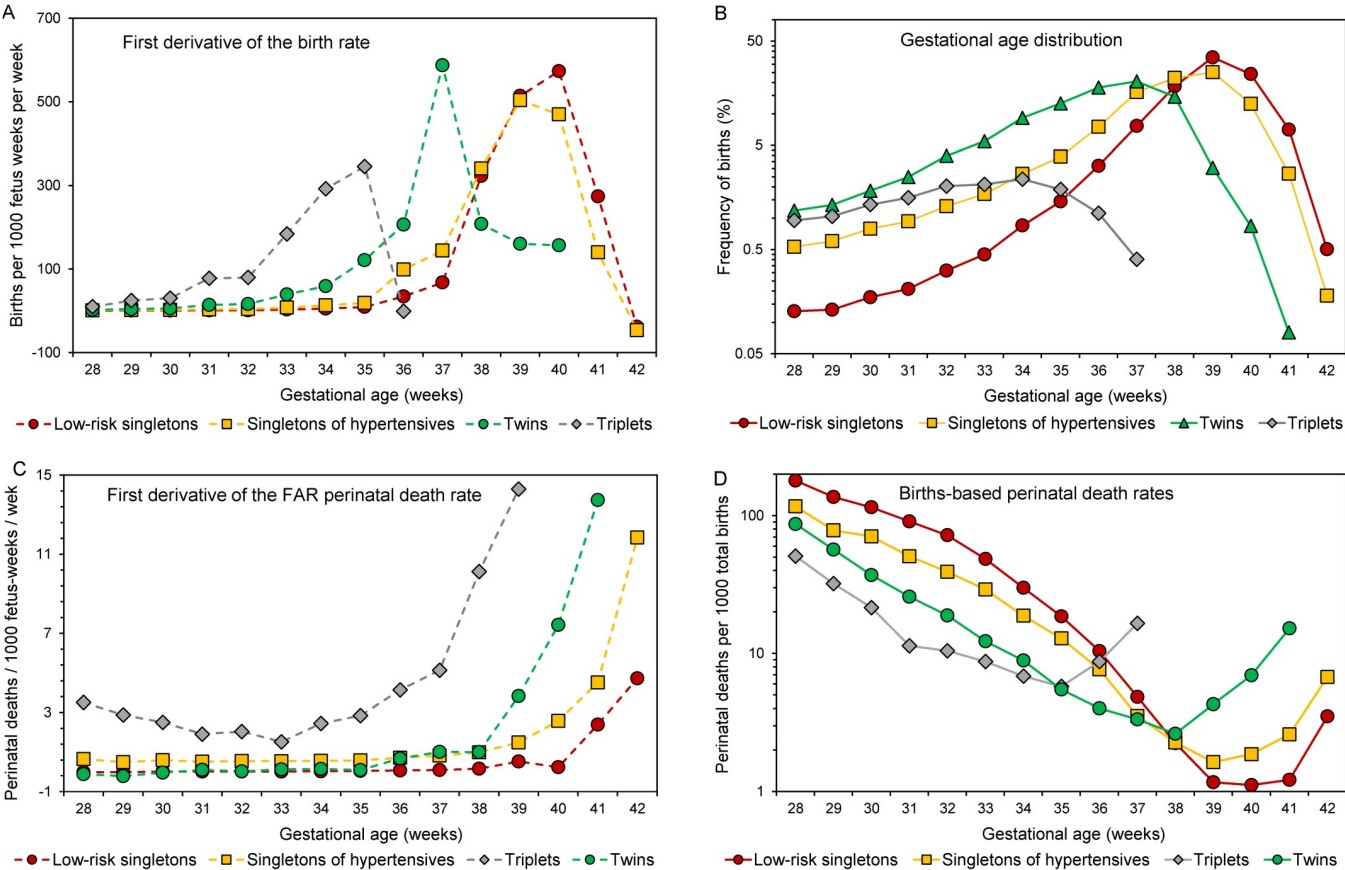

**Fig 4.** The first derivative of the birth rate (Panel A), the gestational age distribution (Panel B), the first derivative of the fetuses-at-risk perinatal death rate (Panel C), and births-based gestational age-specific perinatal death rates (Panel D), among 4 low- and high-risk cohorts, United States, 2004–2015.

derivatives of the birth rate (Fig 3A) and the fetuses-at-risk perinatal death rate (Fig 3B) were left-shifted among twins and a corresponding inverse pattern and left-shift was evident in the births-based gestational age-specific perinatal death rates of twins (Fig 3C).

Fig 4 presents the first derivative of the birth rate (panel A), the gestational age distribution (panel B), the first derivative of the fetuses-at-risk perinatal death rate (panel C) and the births-based perinatal death rate (panel D) among low-risk singletons, singletons of women with hypertension, twins and triplets. The higher-risk cohorts showed a *markedly increasing left-shift* in the pattern of each of these indices compared with the same pattern among the lower-risk cohorts, and birth-based perinatal death rates at optimal gestational age were higher in the higher-risk cohorts. Similar patterns were evident in the first derivatives of the fetuses-at-risk birth rate and perinatal mortality rate, the gestational age distribution and births-based gestational age-specific perinatal death rates of other cohorts (S2-S5 Figs in S1 Appendix).

Fig 5 and Table 1 show that the gestational week at which the first derivative of the birth rate peaked was positively correlated (clustered together) with the mean, mode, and median of gestational age in the 12 low- and high-risk cohorts. The peak in the first derivative of the birth rate was also positively correlated with optimal gestational age and the gestational age at which the first derivative of the fetuses-at-risk perinatal death rate showed a sharp increase in late gestation (Table 1). On the other hand, there was a significant inverse

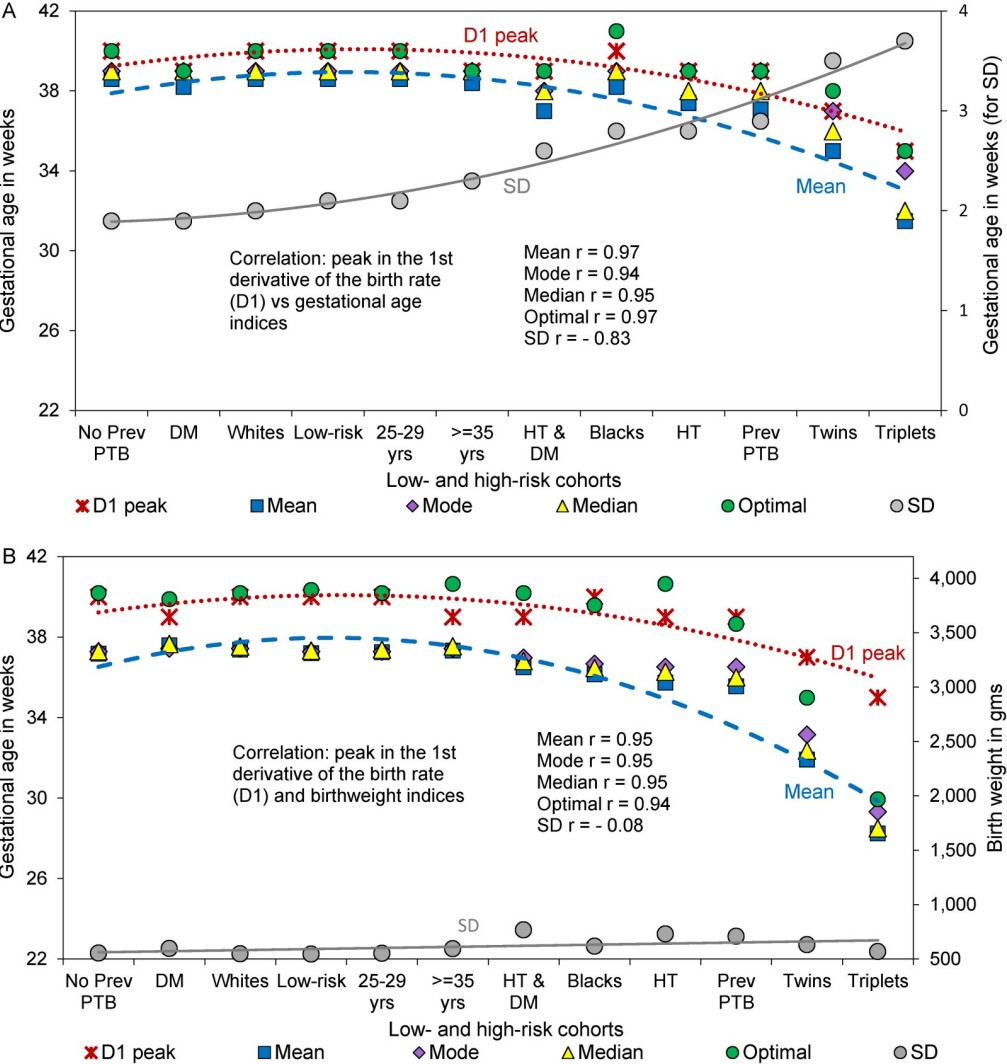

**Fig 5.** Clustering and correlation between the gestational age peak in the first derivative of the birth rate and the mean, mode and median of the gestational age distribution, optimal gestational age, and the standard deviation of the gestational age distribution (Panel A); and clustering and correlation between the gestational age peak in the first derivative of the birth rate and the mean, mode and median birthweight, optimal birthweight, and the standard deviation of the birthweight distribution (Panel B) among 12 low- and high-risk cohorts, United States, 2004–2015. (Cohort notations: No prev PTB denotes no previous preterm birth; DM, diabetes mellitus; Whites, White women; No HT-DM, singletons of women without hypertension or diabetes; 25–29 yrs, women 25–29 years of age; ≥35 yrs, women ≥35 years of age; HT & DM, hypertension and diabetes; Blacks, black women; HT, hypertension; and Prev PTB, previous preterm birth. Note: All series in Panel A are represented on the primary Y-axis except the SD of the gestational age distribution, which is represented on the secondary Y-axis. In Panel B, the D1 peak is represented on the primary Y-axis and all other series are on the secondary Y-axis).

correlation between the peak in the first derivative of the birth rate and the standard deviation of gestation age, and no significant correlation between the peak in the first derivative of the birth rate and the standard deviation of birthweight. The gestational age at which the first derivative of the fetuses-at-risk perinatal death rate showed a sharp increase was positively correlated with optimal gestational age (Table 1) and optimal birthweight (S5 Table in S1 Appendix).

**Table 1. Clustering and correlation between the gestational week at which the first derivative of the birth rate peaked vs the mean, mode, median and standard deviation of the gestational age distribution, optimal gestational age and the gestational week at which the first derivative of the fetuses-at-risk perinatal death rate increased sharply, low- and high-risk cohorts, United States, 2004–2015.**

| Cohort | Peak in the 1st derivative of the birth rate (weeks) | Gestational age distribution | | | | | Sharp increase in 1st derivative of FAR perinatal death rate (weeks) |
|---|---|---|---|---|---|---|---|
| | | Mean | SD | Mode | Median | Optimal | |
| Singletons of women–No HT or DM | 40 | 38.6 | 2.1 | 39 | 39 | 40 | 40 |
| Singletons of HT women | 39 | 37.4 | 2.8 | 39 | 38 | 39 | 41 |
| Singletons of DM women | 39 | 38.2 | 1.9 | 39 | 39 | 39 | 41 |
| Singletons of HT and DM women | 39 | 37.0 | 2.6 | 38 | 38 | 39 | 39 |
| Twins | 37 | 35.0 | 3.5 | 37 | 36 | 38 | 38 |
| Triplets | 35 | 31.5 | 3.7 | 34 | 32 | 35 | 37 |
| Younger mother (25–29 years) | 40 | 38.6 | 2.1 | 39 | 39 | 40 | 41 |
| Older mothers (≥35 years) | 39 | 38.4 | 2.3 | 39 | 39 | 39 | 40 |
| Whites | 40 | 38.6 | 2.0 | 39 | 39 | 40 | 40 |
| Blacks | 40 | 38.2 | 2.8 | 39 | 39 | 41 | 40 |
| Previous preterm birth | 39 | 37.1 | 2.9 | 39 | 38 | 39 | 41 |
| No previous preterm birth | 40 | 38.6 | 1.9 | 39 | 39 | 40 | 40 |
| Pearson r (with D1 peak of birth rate)[a] | 1.00 | 0.97 | -0.83 | 0.94 | 0.96 | 0.97 | 0.86 |
| P value | - | <0.001 | <0.001 | <0.001 | <0.001 | <0.001 | <0.001 |
| 95% confidence interval | - | 0.90, 0.99 | -0.95, -0.48 | 0.80, 0.98 | 0.87, 0.99 | 0.88, 0.99 | 0.58, 0.96 |
| Pearson r (with D1 inflection of FAR perinatal death rate)[b] | | | | | | 0.82 | 1.00 |
| P value | | | | | | <0.001 | - |
| 95% confidence interval | | | | | | 0.46, 0.95 | - |

D1 denotes the first derivative; SD standard deviation; FAR fetuses at risk; HT hypertension; and DM diabetes mellitus.

Optimal gestational age refers to the point in the gestational age distribution at which the perinatal death rate is lowest (see text).

[a] Pearson correlation between the gestational week at which the first derivative (D1) of the birth rate peaks and other indices (n = 12).

[b] Pearson correlation between the gestational week at which the first derivative (D1) of the fetuses-at-risk perinatal death rate increases sharply and optimal gestational age (n = 12).

## Discussion

This study confirms that the gestational age distribution and modal gestational age are determined by the rate of change in the birth rate, while the births-based gestational age-specific perinatal death rate pattern and optimal gestational age are influenced by both the rate of change in the birth rate and the rate of change in the fetuses-at-risk perinatal death rate [29,30]. Also, the lowest perinatal death rate in any cohort, achieved at optimal gestational age, occurs earlier in gestation and is higher in higher-risk cohorts compared with lower-risk cohorts. Lastly, there is a clustering and correlation between the mean, mode and median of gestational age and the optimal gestational age of a cohort, and a symmetric left-shift in births-based gestational age-specific perinatal death rates among higher-risk cohorts. The singular influence on modal gestational age, the dual influences on optimal gestational age, the earlier and higher optimal gestational age in higher-risk cohorts, the clustering and correlation between modal and optimal gestational age, and the symmetrical left-shift in births-based gestational age-specific perinatal death rates among higher-risk cohorts explain the relationships

between modal and optimal gestational age and the mechanism by which relative gestational age resolves the paradox of intersecting perinatal mortality curves (see below).

## Why is modal gestational age typically less than optimal gestational age?

The acceleration in the birth rate peaks earlier in higher-risk cohorts compared with lower-risk cohorts (e.g., at 35, 37, 39 and 40 weeks' gestation, respectively, among triplets, twins, singletons of women with hypertension, and low-risk singletons; Fig 4A) and this influences the gestational age distribution and modal gestational age (34, 37, 39 and 39 weeks, respectively, among the same four cohorts; Fig 4B). It has been suggested that the greater acceleration in the birth rate of higher-risk cohorts represents an exaggerated, hypersensitivity-type response to adverse influences in pregnancy, and could reflect an evolutionary mechanism that prioritises maternal survival in the face of potential threats to fetal well-being [29]. However, the mechanism underlying peak acceleration in the birth rate, and its subsequent decline is unclear and one postulated explanation involves a depletion of susceptibles: pregnancies that reach late gestation are less responsive to hormonal and other triggers that initiate parturition [29].

The births-based perinatal death rate pattern, on the other hand, is influenced by both the rate of change in the birth rate and also by the rate of change in the fetuses-at-risk perinatal death rate. The latter increases abruptly in late gestation (e.g., at 37, 38, 41 and 40 weeks among triplets, twins, singletons of women with hypertension, and low-risk singletons, respectively; Fig 4C) and this ensures that optimal gestational age (35, 38, 39 weeks and 40 weeks, respectively among the four cohorts; Fig 4D) typically exceeds modal gestational age. Similar relationships ensure that optimal birthweight exceeds modal birthweight.

## How does relative gestational age resolve the intersecting mortality curves paradox?

The perinatal death rate achieved at optimal gestational age is higher in higher-risk cohorts compared with lower-risk cohorts. Also, there is clustering together and correlation between the peak in the first derivative of the birth rate and a) the mean, mode and median of the gestational age distribution; and b) optimal gestational age. Optimal gestational age is also correlated with the gestational age at which the first derivative of the fetuses-at-risk perinatal death rate increases in late gestation. These positive correlations mean that a left-shift in the peak of the first derivative of the birth rate will result in a lower modal gestational age, and that left-shifts in the first derivatives of the birth rate and the fetuses-at-risk perinatal death rate will result in a lower optimal gestational age. These features ensure that the (absolute) births-based perinatal mortality rate at modal gestation is higher in higher-risk cohorts than in lower-risk cohorts. In fact, the symmetric left-shift in births-based perinatal death rates in higher-risk cohorts ensures that relative gestational age-specific and relative birthweight-specific perinatal death rates are higher in higher-risk cohorts at all gestational ages and birthweights.

## Strengths

The empirical patterns in this study were based on a large perinatal dataset that permitted examination of several low- and high-risk cohorts. First derivatives of fetuses-at-risk birth and perinatal death rates were calculated to provide insight into mechanisms by which changes in these rates influenced gestational age distributions and births-based perinatal mortality patterns of diverse populations. The use of first derivatives in this context is appropriate because the exponentially rising birth rate conceals large differences in the rate of change in the birth rate between early and later gestation.

## Limitations

The study population was restricted to births 20–43 week's gestation and pregnancy losses that occurred prior to 20 weeks were not included in the study's fetuses-at-risk denominators. Further, the period linked births-infant deaths files used for the study were essentially cross-sectional in nature (unlike the cohort linked births-infant death files). Although all gestational age information in the study was based on the more reliable clinical estimate of gestation, some errors in gestational age were inevitable. Also, the data source provided gestational age by week and not days, and this imprecision likely resulted in small inaccuracies in the indices estimated.

Gestational age-specific fetal growth-restriction rates could not be incorporated into the models because such information was not available. Modeling perinatal mortality using a comprehensive framework incorporating birth and growth- restriction has to await empirical data on gestational age-specific fetal growth-restriction (since revealed growth-restriction patterns [35] only provide an approximation that is influenced by birth rates). Additionally, the analyses presented did not incorporate obstetric intervention (through labour induction and cesarean delivery) which would have impacted gestational age and gestational age-specific perinatal mortality rates. This influence is likely to have affected several indices, especially the gestational age at which the fetuses-at-risk perinatal death rate showed a sharp increase in late gestation (although the inter-relationships between indices was likely unaffected). Thus, changes in obstetric and neonatal care, which have impacted birth and perinatal mortality rates over recent decades, likely did not compromise the relative gestational age-specific analyses in this study as contrasted cohorts (e.g., singletons of low-risk women vs twins) would have been affected almost uniformly by period and cohort effects.

Another weakness of the study was the non-independent nature of the observations: analyses did not account for births to the same woman, and the 12 cohorts studied were not all independent. Although this would have affected variance estimates and P values, inferences based on this large dataset are unlikely to have been seriously compromised. Finally, the validity of the birth data used in this study is low with regard to maternal medical diagnoses such as diabetes/hypertension and previous preterm birth [37,38]. Nevertheless, these medical factors distinguished low- and higher-risk cohorts, provided substantial variability in gestational age distributions and perinatal mortality rate patterns, and illustrated modal, optimal and relative gestational age.

## Interpretation and conclusions

The left-shift in the distribution of the first derivative of the birth rate in higher-risk cohorts results in a symmetrical left-shift in the gestational age distribution and an inversely symmetrical left-shift in the births-based gestational age-specific perinatal death rate curve. Evidence for the added influence of first derivative of the fetuses-at-risk perinatal death rate (left-shifted in higher-risk cohorts) on the births-based perinatal death rate pattern comes from optimal gestational age typically lagging modal gestational age. The symmetric left-shift in the gestational age distribution and the symmetric and inverse left-shift in the births-based perinatal death rate in higher-risk cohorts ensures that relative gestational age-and relative birthweight-specific perinatal death rates are higher among higher-risk cohorts at all gestational ages and birthweights.

The structure and symmetry of gestational age- and birthweight-specific perinatal phenomena provide a powerful narrative: the pattern of the first derivative of the birth rate is congruent with the shape of the gestational age distribution in low- and high-risk cohorts, and there is a compelling inverse symmetry between the pattern of the first derivative of the birth rate

and the births-based perinatal death rate curve. Such symmetry may invoke the concept of epidemiologic beauty, though it should be noted that in modern physics, beauty is regarded by some as a characteristic of nature and by others as an ill-conceived aesthetic bias that has led physics astray. Irrespective of whether or not one finds the symmetry appealing, these explanations provide insight into several birth-based phenomena that have previously defied resolution.

## Supporting information

**S1 Appendix. Contains S1-S5 Fig and S1-S5 Table. S1 Figure**: Birthweight distribution in grams between 3,000 and 3,500 gms, United States, 2004–2015. **S2 Figure**: Contrasts of indices of interest, singletons of women 25–29 vs ≥35 years of age, United States, 2004–15. **S3 Figure**: Contrasts of indices of interest, singletons of White women vs Black women, United States, 2004–15. **S4 Figure**: Contrasts of indices of interest, singletons of low-risk women (i.e., without hypertension or diabetes) vs singletons of women with hypertension and diabetes, United States, 2004–15. **S5 Figure**: Contrasts of indices of interest, singletons of Black women (Panel A) and singletons of women with hypertension, United States, 2004–15. **S1 Table**: Numbers and rates of births and perinatal deaths among singletons of women without hypertension or diabetes, United States, 2004–2015. **S2 Table**: Numbers and rates of births and perinatal deaths among twins, United States, 2004–2015. **S3 Table**: SAS code for quantifying the first and second derivatives of the birth rate. **S4 Table**: Numbers of total births and perinatal deaths, and perinatal death rates in low- and high-risk cohorts, United States, 2004–2015. **S5 Table**: Correlation between gestational and birthweight indices, low- and high-risk cohorts, United States, 2004–2015.
(PDF)

## Author Contributions

**Conceptualization:** K. S. Joseph.

**Formal analysis:** K. S. Joseph.

**Methodology:** K. S. Joseph.

**Visualization:** K. S. Joseph.

**Writing – original draft:** K. S. Joseph.

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
