## [Decision Letter · Decision Letter 0]

29 Sep 2020

PONE-D-20-25958

A compelling symmetry: The extended fetuses-at-risk perspective on modal, optimal and relative birthweight and gestational age

PLOS ONE

Dear Dr. Joseph,

Thank you for submitting your manuscript to PLOS ONE. After careful consideration, we feel that it has merit but does not fully meet PLOS ONE’s publication criteria as it currently stands. Therefore, we invite you to submit a revised version of the manuscript that addresses the points raised during the review process.

ALL of the reviewer's comments must be addressed in your revised manuscript.

We look forward to receiving your revised manuscript.

Kind regards,

Frank T. Spradley

Academic Editor

PLOS ONE

Reviewers' comments:

Reviewer's Responses to Questions

**Comments to the Author**

1. Is the manuscript technically sound, and do the data support the conclusions?

Reviewer #1: Yes

Reviewer #2: Yes

2. Has the statistical analysis been performed appropriately and rigorously? 

Reviewer #1: Yes

Reviewer #2: Yes

3. Have the authors made all data underlying the findings in their manuscript fully available?

Reviewer #1: Yes

Reviewer #2: Yes

4. Is the manuscript presented in an intelligible fashion and written in standard English?

Reviewer #1: Yes

Reviewer #2: Yes

5. Review Comments to the Author

Reviewer #1: The research article is informative but diagrams would look better if multiple colors could be used.

Please explain why acceleration in the birth rate peaks earlier in higher-risk cohorts compared with lower-risk cohorts

Reviewer #2: This manuscript presents further analysis on a larger data set from NCHS birth-death linkage records with the purpose of presenting an analysis of perinatal deaths taking into account the change of such deaths by week of gestation.

The authors obtained an estimate on change of rates from cubic spline or polynomial curve fitting of the rates using the number of fetuses at risk, and describes an assessment of perinatal deaths taking into account differences in birth rates by gestational age. The approach proposed by the authors is similar to the Bongaarts and Feeney approach to demographic phenomena and it was received with skepticism and not widely adopted to the assessment of cross-sectional data such as those presented here. The CDC NVSS calls the data birth cohort, but they are in nature cross-sectional data, as opposed to true birth cohort studies that collect information as early as possible during pregnancy. The authors may want to address this limitation of their data. An example of a birth cohort study can be found in the reference below.

1. Olsen J, Melbye M, Olsen SF, Sørensen TI, Aaby P, Andersen AM, Taxbøl D, Hansen KD, Juhl M, Schow TB, Sørensen HT, Andresen J, Mortensen EL, Olesen AW, Søndergaard C. The Danish National Birth Cohort--its background, structure and aim. Scand J Public Health. 2001 Dec;29(4):300-7. doi: 10.1177/14034948010290040201. PMID: 11775787.

6. PLOS authors have the option to publish the peer review history of their article (what does this mean?). If published, this will include your full peer review and any attached files.

Reviewer #1: **Yes: **Farzana Ahmed

Reviewer #2: **Yes: **Victor M Cardenas

---

## [Author Response · Author response to Decision Letter 0]

7 Oct 2020

Reviewer #1:

Comment: The research article is informative but diagrams would look better if multiple colors could be used.

Response: The Figures have been revised and colors have been used to distinguish the curves in the graphs.

Comment: Please explain why acceleration in the birth rate peaks earlier in higher-risk cohorts compared with lower-risk cohorts.

Response: The revised manuscript now includes an explanation for why the birth rate peaks earlier in higher risk cohorts (Page 13-14 lines 285-91)

“It has been suggested that the greater acceleration in the birth rate of higher-risk cohorts represents an exaggerated, hypersensitivity‐type response to adverse influences in pregnancy, and could reflect an evolutionary mechanism that prioritises maternal survival in the face of potential threats to fetal well‐being [29]. However, the mechanism underlying peak acceleration in the birth rate, and its subsequent decline is unclear and one postulated explanation involves a depletion of susceptibles: pregnancies that reach late gestation are less responsive to hormonal and other triggers that initiate parturition [29].”

Reviewer #2:

Comment: This manuscript presents further analysis on a larger data set from NCHS birth-death linkage records with the purpose of presenting an analysis of perinatal deaths taking into account the change of such deaths by week of gestation.

The authors obtained an estimate on change of rates from cubic spline or polynomial curve fitting of the rates using the number of fetuses at risk, and describes an assessment of perinatal deaths taking into account differences in birth rates by gestational age. The approach proposed by the authors is similar to the Bongaarts and Feeney approach to demographic phenomena and it was received with skepticism and not widely adopted to the assessment of cross-sectional data such as those presented here. The CDC NVSS calls the data birth cohort, but they are in nature cross-sectional data, as opposed to true birth cohort studies that collect information as early as possible during pregnancy. The authors may want to address this limitation of their data. An example of a birth cohort study can be found in the reference below.

1. Olsen J, Melbye M, Olsen SF, Sørensen TI, Aaby P, Andersen AM, Taxbøl D, Hansen KD, Juhl M, Schow TB, Sørensen HT, Andresen J, Mortensen EL, Olesen AW, Søndergaard C. The Danish National Birth Cohort--its background, structure and aim. Scand J Public Health. 2001 Dec;29(4):300-7. doi: 10.1177/14034948010290040201. PMID: 11775787.

Response: The data used for the study was obtained from the period linked birth-infant death data files of the CDC’s National Center for Health Statistics (available at https://www.cdc.gov /nchs/data_access/vitalstatsonline.htm). The period linked birth-infant death files are different from the NVSS birth files (also available in the same data repository). The distinction between these files arises because the period linked birth-infant death files contain linked information from the birth file and the (fetal and infant) death files. For instance, the NVSS birth file for 2017, includes information on all live births that occurred in the United States in 2017. On the other hand, the 2017 period linked birth-infant death file includes information from the

a) NVSS births file with information on all live births that occurred in 2017

b) fetal death file which includes information on all stillbirths that occurred in 2017

c) death file with information on all infants deaths that occurred in 2017

The 3 files are linked with the intent of documenting the longitudinal experience of all viable births in 2017, with the events of interest restricted to fetal deaths that occur after 20 weeks’ gestation, and infant deaths i.e., deaths that occur within 1 year after birth. Since all live births and stillbirths that occur in 2017 are included in the file, this file does constitute the full 2017 birth cohort denominator. The longitudinal follow up for infant deaths that occur in 2018 to live births occurring in 2017 is not included; this is approximated by including infant deaths in 2017 among live births that occurred in 2016. Note that the cohort birth-infant death files for 2017 include the longitudinal follow up into 2018 and hence availability of these files is delayed by 1 year. In short, the period linked birth-infant death files document the longitudinal experience of a birth cohort’s experience with regard to perinatal mortality and infant mortality.

The period and cohort effects to which the Reviewer alludes are well understood within the perinatal epidemiology community and are generally referred to under the concept termed ‘age-period-cohort’ effects. This paper deals primarily with (gestational) age-specific effects among different groups/cohort and period and cohort effects are not explored. Period effects and cohort effects would have shaped some of the phenomena explored in the study: increases in iatrogenic early delivery have increased late preterm birth in recent decades, and obstetric and neonatal care has substantially reduced perinatal death rates. Cohort effects in terms of alterations to maternal health have probably been less obvious, though general improvements in maternal health in recent cohorts and increases in the fertility of women with chronic disease have likely played a role in shaping perinatal mortality trends. However, these phenomena are not likely to have a material impact in shaping the relative (gestational) age-specific patterns of birth and perinatal death rates among contrasted groups e.g., low-risk singletons vs singletons of mothers with hypertension since all contrasts of intersecting mortality curves were within a given period and cohort. The revised manuscript include a sentence regarding this issue in the limitations section of the manuscript (Page 16, line 334-37).

“Thus, changes in obstetric and neonatal care, which have impacted birth and perinatal mortality rates over recent decades, likely did not compromise the relative gestational age-specific analyses in this study as contrasted cohorts (e.g., singletons of low-risk women vs twins) would have been affected almost uniformly by period and cohort effects.”

---

## [Decision Letter · Decision Letter 1]

6 Nov 2020

PONE-D-20-25958R1

A compelling symmetry: The extended fetuses-at-risk perspective on modal, optimal and relative birthweight and gestational age

PLOS ONE

Dear Dr. Joseph,

Thank you for submitting your manuscript to PLOS ONE. After careful consideration, we feel that it has merit but does not fully meet PLOS ONE’s publication criteria as it currently stands. Therefore, we invite you to submit a revised version of the manuscript that addresses the points raised during the review process.

There is still a comment that should be addressed.

We look forward to receiving your revised manuscript.

Kind regards,

Frank T. Spradley

Academic Editor

PLOS ONE

Reviewers' comments:

Reviewer's Responses to Questions

**Comments to the Author**

1. If the authors have adequately addressed your comments raised in a previous round of review and you feel that this manuscript is now acceptable for publication, you may indicate that here to bypass the “Comments to the Author” section, enter your conflict of interest statement in the “Confidential to Editor” section, and submit your "Accept" recommendation.

Reviewer #1: All comments have been addressed

Reviewer #2: (No Response)

2. Is the manuscript technically sound, and do the data support the conclusions?

Reviewer #1: Yes

Reviewer #2: (No Response)

3. Has the statistical analysis been performed appropriately and rigorously? 

Reviewer #1: Yes

Reviewer #2: (No Response)

4. Have the authors made all data underlying the findings in their manuscript fully available?

Reviewer #1: Yes

Reviewer #2: (No Response)

5. Is the manuscript presented in an intelligible fashion and written in standard English?

Reviewer #1: Yes

Reviewer #2: (No Response)

6. Review Comments to the Author

Reviewer #1: (No Response)

Reviewer #2: The author should acknowledge the limitation of the data to deaths occurring <20 weeks of gestation (about 20% of pregnancies end up in pregnancy losses, the births may not truly represent the denominator (persons at-risk). The data is birth-death linkage cross-sectional in nature not from a cohort study.

7. PLOS authors have the option to publish the peer review history of their article (what does this mean?). If published, this will include your full peer review and any attached files.

Reviewer #1: **Yes: **Farzana Ahmed

Reviewer #2: **Yes: **Victor M Cardenas

---

## [Author Response · Author response to Decision Letter 1]

9 Nov 2020

Response to Reviewers’ comments

Comments to the Author

1. If the authors have adequately addressed your comments raised in a previous round of review and you feel that this manuscript is now acceptable for publication, you may indicate that here to bypass the “Comments to the Author” section, enter your conflict of interest statement in the “Confidential to Editor” section, and submit your "Accept" recommendation.

Reviewer #1: All comments have been addressed

Reviewer #2: (No Response)

Response: No response required.

 2. Is the manuscript technically sound, and do the data support the conclusions?

Reviewer #1: Yes

Reviewer #2: (No Response)

Response: No response required.

3. Has the statistical analysis been performed appropriately and rigorously? 

 Reviewer #1: Yes

Reviewer #2: (No Response)

Response: No response required.

 4. Have the authors made all data underlying the findings in their manuscript fully available?

Reviewer #1: Yes

Reviewer #2: (No Response)

Response: No response required.

5. Is the manuscript presented in an intelligible fashion and written in standard English?

Reviewer #1: Yes

Reviewer #2: (No Response)

Response: No response required.

6. Review Comments to the Author

Reviewer #1: (No Response)

Reviewer #2: The author should acknowledge the limitation of the data to deaths occurring <20 weeks of gestation (about 20% of pregnancies end up in pregnancy losses, the births may not truly represent the denominator (persons at-risk). The data is birth-death linkage cross-sectional in nature not from a cohort study.

Response: A sentence has been added to the Limitations section of the manuscript which acknowledges that the study population was restricted to births at and above 20 weeks’ gestation and that pregnancy losses occurring before 20 weeks were not included in the study’s fetuses-at-risk denominators. Additionally, the revised manuscript includes a sentence stating that the period linked births-infant deaths files are essentially cross sectional in nature (unlike the cohort linked births-infant death files). 

Page 15, line 322-325

“The study population was restricted to births 20-43 week’s gestation and pregnancy losses that occurred prior to 20 weeks were not included in the study’s fetuses-at-risk denominators. Further, the period linked births-infant deaths files used for the study are essentially cross-sectional in nature (unlike the cohort linked births-infant death files).”

---

## [Decision Letter · Decision Letter 2]

13 Nov 2020

A compelling symmetry: The extended fetuses-at-risk perspective on modal, optimal and relative birthweight and gestational age

PONE-D-20-25958R2

Dear Dr. Joseph,

We’re pleased to inform you that your manuscript has been judged scientifically suitable for publication and will be formally accepted for publication once it meets all outstanding technical requirements.

Kind regards,

Frank T. Spradley

Academic Editor

PLOS ONE

Reviewers' comments:

Reviewer's Responses to Questions

**Comments to the Author**

1. If the authors have adequately addressed your comments raised in a previous round of review and you feel that this manuscript is now acceptable for publication, you may indicate that here to bypass the “Comments to the Author” section, enter your conflict of interest statement in the “Confidential to Editor” section, and submit your "Accept" recommendation.

Reviewer #2: (No Response)

2. Is the manuscript technically sound, and do the data support the conclusions?

Reviewer #2: Yes

3. Has the statistical analysis been performed appropriately and rigorously? 

Reviewer #2: Yes

4. Have the authors made all data underlying the findings in their manuscript fully available?

Reviewer #2: Yes

5. Is the manuscript presented in an intelligible fashion and written in standard English?

Reviewer #2: Yes

6. Review Comments to the Author

Reviewer #2: (No Response)

7. PLOS authors have the option to publish the peer review history of their article (what does this mean?). If published, this will include your full peer review and any attached files.

Reviewer #2: No

---

## [Editor Report · Acceptance letter]

17 Nov 2020

PONE-D-20-25958R2 

A compelling symmetry: The extended fetuses-at-risk perspective on modal, optimal and relative birthweight and gestational age 

Dear Dr. Joseph:

I'm pleased to inform you that your manuscript has been deemed suitable for publication in PLOS ONE. Congratulations! Your manuscript is now with our production department. 

Kind regards, 

on behalf of

Dr. Frank T. Spradley 

Academic Editor

PLOS ONE